# Genetic landscape of prostate cancer conspicuity on multiparametric MRI: a protocol for a systematic review and bioinformatic analysis

Joseph M Norris [ID],[1] Benjamin S Simpson [ID],[1] Marina A Parry,[2] Clare Allen,[3] Rhys Ball,[4] Alex Freeman,[4] Daniel Kelly,[5] Alex Kirkham,[3] Veeru Kasivisvanathan [ID],[1] Hayley C Whitaker,[1] Mark Emberton[1]

JMN and BSS are joint first authors.

For numbered affiliations see end of article.

**Correspondence to**
Joseph M Norris;
joseph.norris@ucl.ac.uk

## ABSTRACT

**Introduction** The introduction of multiparametric MRI (mpMRI) has enabled enhanced risk stratification for men at risk of prostate cancer, through accurate prebiopsy identification of clinically significant disease. However, approximately 10%–20% of significant prostate cancer may be missed on mpMRI. It appears that the genomic basis of lesion visibility or invisibility on mpMRI may have key implications for prognosis and treatment. Here, we describe a protocol for the first systematic review and novel bioinformatic analysis of the genomic basis of prostate cancer conspicuity on mpMRI.

**Methods and analysis** A systematic search of MEDLINE, PubMed, EMBASE and Cochrane databases will be conducted. Screening, data extraction, statistical analysis and reporting will be performed in accordance with the Preferred Reporting Items for Systematic Reviews and Meta-Analyses (PRISMA) guidelines. Included papers will be full text articles, written between January 1980 and December 2019, comparing molecular characteristics of mpMRI-visible lesions and mpMRI-invisible lesions at the DNA, DNA-methylation, RNA or protein level. Study bias and quality will be assessed using a modified Newcastle-Ottawa score. Additionally, we will conduct a novel bioinformatic analysis of supplementary material and publicly available data, to combine transcriptomic data and reveal common pathways highlighted across studies. To ensure methodological rigour, this protocol is written in accordance with the PRISMA Protocol 2015 checklist.

**Ethics and dissemination** Ethical approval will not be required, as this is an academic review of published literature. Findings will be disseminated through publications in peer-reviewed journals, and presentations at national and international conferences.

**PROSPERO registration number** CRD42019147423.

## Strengths and limitations of this study

► This project will be the first comprehensive review of the molecular characteristics of multiparametric MRI (mpMRI)-visible and mpMRI-invisible lesions, following the key methodological steps of the Preferred Reporting Items for Systematic Reviews and Meta-Analyses guidelines.

► Novel bioinformatic over-representation analysis will reveal shared pathways and genes involved in prostate cancer mpMRI conspicuity that are overlooked by individual studies.

► The degree of heterogeneity between the included studies limits the comparability of derived results.

► There are relatively few studies that directly compare the genomic basis of mpMRI-visible and mpMRI-invisible lesions which potentially limits the strength of conclusions from this work.

significant prostate cancers remain mpMRI invisible.[1 3 4]

Suspicious mpMRI phenotypes appear to closely associate with the presence of prostate cancer, and to an extent, its Gleason grade.[1] However, the degree to which the molecular landscape of prostate cancer correlates with mpMRI is unclear. In contrast, the general molecular basis of prostate cancer is well studied, with large-scale genomic enquires defining key drivers in tumour development and progression. Aberrations, including recurrent mutations in SPOP1, FOXA1, IDH1, fusions in TMPRSS/ERG, ETV1/4, FLI1 and copy number alterations, such as MYC amplification or NKX3-1, RB1 and PTEN deletions or transcriptomic changes in AMACR, PCA3, GDF15 and MSMB, have all been investigated in the context of prostate cancer.[5–7] Over the past 5 years, there has been an increased effort to use this to develop our understanding of the genetic

## BACKGROUND

Prebiopsy multiparametric MRI (mpMRI) has excellent diagnostic accuracy, and has improved risk stratification in prostate cancer diagnosis, compared to systematic transrectal ultrasound-guided prostate biopsy.[1 2] Despite this, approximately 10%–20% of clinically



basis of prostate cancer conspicuity on mpMRI. It is now prudent to collate this growing evidence base, to clarify on progress made so far, and identify the most pertinent areas for future research.

The aim of this systematic review and bioinformatic analysis is to determine and summarise for the first time, the genomic correlates of tumour visibility and invisibility on mpMRI, in order to elucidate the mechanisms that underpin mpMRI conspicuity, and the prognostic implication of mpMRI phenotypes.

## METHODS AND ANALYSIS

This protocol of the planned systematic review and bioinformatic analysis is written in line with the Preferred Reporting Items for Systematic Reviews and Meta-Analyses Protocol (PRISMA-P) 2015 checklist.[8] Selected studies will be used for thematic synthesis and appropriate supplementary material extracted. In addition to the literature search, a bioinformatic approach will be used to identify relevant publicly available genomic data for a comprehensive analysis of over-represented pathways and biological functions.

### Search methodology

Searches will be carried out using the MEDLINE, PubMed, EMBASE and Cochrane databases in order to retrieve maximum yield of relevant evidence. The search will include Medical Subject Headings terms and free text, combined with Boolean operators. The search will include the terms: 'prostate', 'cancer' and 'MRI', as well as multiple synonyms for the term 'genetics', to account for large heterogeneity in nomenclature and diversity of topics encompassed within this heading. Identified articles will be uploaded to Rayyan, a semiautomated tool to expedite the initial screening process and to allow two reviewers to filter duplicate studies and screen articles for relevance.[9] Furthermore, the reference section of all included articles will be searched manually to identify missed studies or additional data. Finally, experts will be consulted to identify additional literature. Only articles published between January 1980 and December 2019 will be included in our review. For the bioinformatic analysis, the National Center for Biotechnology Information Gene Expression Omnibus and European Genome-phenome archive will be searched using the same search parameters as those used for the literature search. If sufficient processed data are available, data will be extracted from supplementary to ensure data veracity. In the case of missing data, the corresponding authors will be contacted directly.

### Study selection and data extraction

Two researchers will independently screen eligible studies, assessing titles and abstracts for relevance. If considered eligible, the full text will be retrieved and further reviewed for eligibility. Any lack of concordance between reviewers will be discussed until a consensus is

| **Table 1** | Data collection items | |
|---|---|---|
| **Item no** | **Data title** | **Data type** |
| 1 | Year of publication | Study characteristic |
| 2 | Study authors | Study characteristic |
| 3 | Study design | Study characteristic |
| 4 | Patient population | Demographics |
| 5 | Number of participants | Demographics |
| 6 | mpMRI scoring scheme used | Methodology |
| 7 | Definition for clinically significant disease | Methodology |
| 8 | Definition for lesion visibility and invisibility | Methodology |
| 9 | Sample processing approach | Methodology |
| 10 | Biomolecule studied | Outcome |
| 11 | Differential expression of biomolecule | Outcome |

mpMRI, multiparametric MRI; no, number.

reached or passed to a third reviewer. Additionally, the reason for exclusion will be noted for later analysis. This process will be documented in detail in order to generate the PRISMA flow diagram.

### Inclusion criteria

To be included in the analysis, studies must investigate one or more genomic aspects of the appearance of prostate cancer on mpMRI. Genomic investigation could be at the DNA level, investigating SNPs or somatic alterations, or focused on larger scale alterations, such as copy number changes. Moreover, investigations revealing higher order structures, such as methylation, will also be included. Transcriptomic data analysing RNA expression (coding or non-coding) or microRNA will also be included, as will investigation into protein expression.

### Exclusion criteria

Conference abstracts, correspondence articles, expert opinions and case reports will be excluded. Studies that do not correlate mpMRI phenotypes with genomic data will be excluded. Articles focusing solely on macro-characteristics or clinical features of mpMRI conspicuity will be removed.

### Data extraction

All extracted data will be collated in a dedicated data sheet. Both reviewers will extract data independently from each other and agree on consensus. Table 1 summarises data items to be collected.

### Endpoints

The primary endpoint will be differential gene expression between mpMRI-visible and mpMRI-invisible lesions. Initial literature searching suggests that the majority of

studies have focused on mRNA expression; therefore, we will identify recurring genes which have a significant fold-change between visible and invisible lesions. Secondary endpoints will include explanatory links between gene function and mpMRI conspicuity, and the prognostic value of differential gene enrichment. We will identify key themes within the literature, with a focus on the MRI scoring systems used (eg, Prostate Imaging–Reporting and Data System (PI-RADS), Likert, radiogenomic features), the criteria used to define lesion visibility (PI-RADS or Likert score thresholds) and the type of cohort used in the study (eg, radical vs biopsy cohort). Additionally, we will identify which studies corrected for histopathological features such as lesion size and Gleason grade.

### Subgroup analysis

A secondary analysis will look at whether gene panels, or individual genes, associated with mpMRI conspicuity show different expression levels when key attributes (eg, Gleason grade and tumour size) are matched between study arms.

### Statistical analysis

For the bioinformatic analysis, we will aim to keep data as close to the original format that was provided in the selected study. In the case of transcriptomic data, we will extract Log2 fold change and the associated false discovery rate adjusted value between mpMRI-visible and mpMRI-invisible tumours. In more complex comparisons, we will attempt to extract details which indicate concordance of differential gene expression between studies; however, we will not include effect size if this is incomparable. If unavailable, we will simply compare highlighted genomic features and the direction of change between groups. Genes highlighted in multiple studies will be used (via over-representation analysis) to identify enriched pathways. This analysis will be performed using the WebGestalt: Gene Set Analysis Toolkit.[10] This method does not rely on effect size weighting and uses a modified Fisher's exact test to identify enriched biological processes.[11] Steps in the bioinformatic analysis are outlined in table 2.

| Table 2 | Steps in the bioinformatic analysis |
|---------|-------------------------------------|
| **Step no** | **Task** |
| Step 1 | Identifying studies with suitable supplementary data or associated data in data repositories. |
| Step 2 | Assessing comparability of results. |
| Step 3 | Comparing overlapping genes in multiple studies. |
| Step 4 | Over-representation analysis of genes present in multiple studies. |
| Step 5 | Comparison of suggested gene panels in independent cohort datasets. |

no, number.

### Risk of bias in individual studies

To assess bias and quality across the included studies, a modified Newcastle-Ottawa score will be used, designed to assess observational cohort studies.[12] This scoring system is split into three main sections: selection, comparability and outcome. Each section contains subquestions that assess the quality of the research methodology, at the study level. Two reviewers will be involved with this process, and any disagreement will be settled by consensus. The results of the bias assessment will be used to influence data synthesis by providing an assessment of the reliability and applicability of the data produced. If studies are found to be of low quality (or high bias) then they may be excluded, or if included, they will be accompanied by appropriate commentary in the discussion. As this review centres around genomic studies of conspicuity as opposed to treatment outcome, non-applicable sections may be modified to reflect the evidence base and reduce reporting inaccuracy.

### Patient and public involvement

Patients and the public were not involved in the development of this systematic review protocol.

## DISCUSSION

In the UK, the National Institute for Health and Care Excellence have recently updated their guidelines for men with suspected prostate cancer.[13] Men with suspected prostate cancer should now undergo mpMRI before prostate biopsy, as part of their risk stratification.[14] This move will likely see a rise in the numbers of mpMRI performed, and as such, we should strive to better understand the nature of disease detected and missed by this technology. Through systematic review and bioinformatic analysis, we aim to identify commonality between genomic-based studies which have investigated mpMRI-visible and mpMRI-invisible tumours. Our results will enhance the understanding of the genomic basis of prostate cancer visibility on mpMRI, the pathways that underpin this and, ultimately, the prognostic implication of disease conspicuity.

In our thematic analyses, we aim to derive the most prominent themes from studies that compare genetic differences in mpMRI-visible and mpMRI-invisible disease. The mechanistic links between genomic characteristics and the appearance of prostate cancer on mpMRI are not clear; however, early evidence suggests that genes controlling tumour cell proliferation may impact on perfusion, and as such, generation of mpMRI signal, particularly on diffusion-weighted imaging sequences.[15] Furthermore, combining suitable studies that link tumour conspicuity to genetic control will strengthen our understanding. Another important potential theme that we aim to elicit is the prognostic implication of different mpMRI-visibility or mpMRI-invisibility phenotypes (as defined by PI-RADS or Likert scoring schemes).[16] It appears that mpMRI-visible prostate cancer tends to have



higher expression of genes associated with poor prognosis; however, this view is not unanimous throughout the literature, and our rigorous systematic analysis will expose key differences in expression of prognostically significant genes, across different studies.[17 18]

In our bioinformatic analyses, we will compare large, publicly available genomic datasets that are linked to included studies from our systematic review. Described by Weidman and Arrison as 'data exploitation', this approach allows effective use of collective information to obtain new insights.[19] We hope to elucidate commonalities between the strongest molecular differences between mpMRI-visible and mpMRI-invisible tumours. Pooling of data will strengthen association of particular genes with previously described features (eg, molecular pathways and prognostic value). Furthermore, bioinformatic comparison of these datasets may reveal potential key targets for future research, including the development of peripheral biomarkers to identify clinically significant, mpMRI-inconspicuous prostate cancer. A potential limitation may be data heterogeneity, due to differences in definitions for MRI-visibility and methods of identifying genetic determinants. Our analysis will attempt to overcome this issue by focusing on studies with similar definitions of visibility, and restricting our analyses to identified genes and enriched pathways, as opposed to effect sizes.

In summary, this systematic review and novel bioinformatic analysis of publicly available genomic data will collate extant evidence in this emerging field, for the first time. Synthesis of these studies will enhance our current understanding of the role that genetics play in the mpMRI-conspicuity of prostate cancer, and will clarify the most pertinent areas for future research.

## Trial status

- ► Preliminary searches: started.
- ► Piloting of the study selection process: started.
- ► Formal screening: started.
- ► Data extraction: not started.
- ► Risk of bias assessment: not started.
- ► Data analysis: not started.

## Draft of search strategy for MEDLINE, EMBASE, PubMed and Cochrane databases

(prostate AND cancer) AND (gene OR genetic OR genome OR genomic OR transcriptome OR transcriptomic OR epigenetic) AND (magnetic resonance imaging OR MRI)

## ETHICS AND DISSEMINATION

Due to the nature of the study, there are no relevant ethical concerns and informed consent will not be required. The protocol and systematic review will be disseminated via a peer-reviewed journal.

## Author affiliations

[1]UCL Division of Surgery & Interventional Science, University College London, London, UK
[2]UCL Cancer Institute, University College London, London, UK
[3]Department of Radiology, University College London Hospitals NHS Foundation Trust, London, UK
[4]Department of Pathology, University College London Hospitals NHS Foundation Trust, London, UK
[5]School of Healthcare Sciences, Cardiff University, Cardiff, UK

**Contributors** The authors' contribution includes, but is not limited to, the following: JMN and BSS drafted the manuscript and created the study concept. MAP gave advice on genetic aspects. CA, RB, AK, AF, DK, VK, HCW and ME provided supervision and guidance during the study. All authors reviewed and approved the manuscript in its current form.

**Funding** JMN is funded by the Medical Research Council (MRC) on an MRC Clinical Research Training Fellowship (MRC Grant Ref: MR/S00680X/1). BSS receives funding from the Rosetrees Foundation.

**Competing interests** None declared.

**Patient consent for publication** Not required.

**Provenance and peer review** Not commissioned; externally peer reviewed.

**ORCID iDs**
Joseph M Norris http://orcid.org/0000-0003-2294-0303
Benjamin S Simpson http://orcid.org/0000-0003-3685-6110
Veeru Kasivisvanathan http://orcid.org/0000-0002-0832-382X

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
