## [Reviewer comments · BMJ Open]

ARTICLE DETAILS

TITLE (PROVISIONAL)	The genetic landscape of prostate cancer conspicuity on multiparametric MRI: a protocol for a systematic review and bioinformatic analysis
AUTHORS	Norris, Joseph Michael; Simpson, Benjamin Scott S; Parry, Marina A; Allen, Clare; Ball, Rhys; Freeman, Alex; Kelly, Daniel; Kirkham, Alex; Kasivisvanathan, Veeru; Whitaker, Hayley C; Emberton, Mark

VERSION 1 – REVIEW

REVIEWER	Oksana Yaskiv Northwell Health System United States of America
REVIEW RETURNED	23-Oct-2019

GENERAL COMMENTS	Well thought out project that may provide new insights into Prostate Cancer genomics
--

REVIEWER	Tobias Nordström Karolinska Institutet, Sweden
REVIEW RETURNED	30-Oct-2019

GENERAL COMMENTS	This is a timely review assessing the associations between genetic alterations and the MRI visibility of prostate cancer tumours. Have a few minor suggestions that the authors might consider: -MRI visibility is rather schematically defined. Could be of value to define how different Likert/PIRADS score will be handled (e.g. would PIRADS 2 score be considered “visible”). Further, how will lesion size be considered if available?-The methodology seems sound. A bit more thorough section on the bioinformatics section could improve reproducibility/transparency.-There is a risk of data heterogeneity. A discussion on limitations and how these are addressed could be of value.
---

REVIEWER	Sungmin Woo Memorial Sloan Kettering Cancer Center, NY, USA
REVIEW RETURNED	17-Dec-2019

GENERAL COMMENTS	The authors have written a Systematic review protocol dealing with the topic of The Genetic Landscape of Prostate Cancer Conspicuity on Multiparametric MRI.
--

	1. The objectives are clear and the methodology supports how the objective will be achieved, in detail. 2. The protocol clearly defines that it will be performed according to the PRISMA guidelines, which is a prerequisite for performing systematic reviews and meta-analyses. Also, it is appreciated that the authors have already registered their protocol in PROSPERO for reproducibility and clarity. 3. Searching MEDLINE, PubMed, EMBASE, and Cochrane databases are sufficient for identifying literature (at least within the English literature). However, clear definition of search date will be needed to be added. 4. The Newcastle-Ottawa scale is appropriate for risk/bias assessment. 5. The pre-defined data to be extracted and subgroup analysis is appropriate. 6. No grammatical errors. 7. no other comments.
--	---

VERSION 1 – AUTHOR RESPONSE

Reviewer: 1

Reviewer Name: Oksana Yaskiv

Institution and Country: Northwell Health System, United States of America

Please state any competing interests or state 'None declared': None declared

Well thought out project that may provide new insights into Prostate Cancer genomics.

- Author Response: Thank you.

Reviewer: 2

Reviewer Name: Tobias Nordström

Institution and Country: Karolinska Institutet, Sweden

Please state any competing interests or state 'None declared': None declared

The authors would like to sincerely thank all the reviewers for their kind comments and insights.

This is a timely review assessing the associations between genetic alterations and the MRI visibility of prostate cancer tumours. Have a few minor suggestions that the authors might consider:

- Author Response: Thank you.

- MRI visibility is rather schematically defined. Could be of value to define how different Likert/PIRADS score will be handled (e.g. would PIRADS 2 score be considered "visible"). Further, how will lesion size be considered if available?

- Author Response: We agree that the definition of MRI-visibility is a critical factor determining which genetic changes are detected. We have now added extra detail in our 'endpoints' section to describing the criteria used for defining visibility. In terms of comparability, each study which has available data to undergo over-representation analysis will be analysed separately due to the heterogeneity caused by different definitions of visibility. We have added detail describing how we will look at which studies have adjusted their analysis for Gleason grade and lesion size. Finally, we will only compare identified genes rather than the effect sizes, for this reason.

- The methodology seems sound. A bit more thorough section on the bioinformatics section could improve reproducibility/transparency.
- Author Response: We have elaborated on the bioinformatic methods used for the analysis, under 'statistical analysis' including a citation for our primary method. We have now decided on the algorithm we will adopt for the over-expression analysis and referenced the peer-reviewed paper.

- There is a risk of data heterogeneity. A discussion on limitations and how these are addressed could be of value.
- Author Response: We have added a line in our discussion to reflect the limitations of data heterogeneity.

Reviewer: 3

Reviewer Name: Sungmin Woo

Institution and Country: Memorial Sloan Kettering Cancer Center, NY, USA

Please state any competing interests or state 'None declared': None

The authors have written a Systematic review protocol dealing with the topic of The Genetic Landscape of Prostate Cancer Conspicuity on Multiparametric MRI.

- Author Response: Thank you.

1. The objectives are clear and the methodology supports how the objective will be achieved, in detail.

- Author Response: Thank you.

2. The protocol clearly defines that it will be performed according to the PRISMA guidelines, which is a prerequisite for performing systematic reviews and meta-analyses. Also, it is appreciated that the authors have already registered their protocol in PROSPERO for reproducibility and clarity.

- Author Response: Thank you.

3. Searching MEDLINE, PubMed, EMBASE, and Cochrane databases are sufficient for identifying literature (at least within the English literature). However, clear definition of search date will be needed to be added.

- Author Response: We have now defined our search dates to articles published between January 1980 - December 2019. We have added this abstract and main article.

4. The Newcastle-Ottawa scale is appropriate for risk/bias assessment.

- Author Response: Thank you.

5. The pre-defined data to be extracted and subgroup analysis is appropriate.

- Author Response: Thank you.

6. No grammatical errors.

- Author Response: Thank you.

7. No other comments.

- Author Response: Thank you.